# The "Psychologization" of Self-Images: Parents Views on the Gendered Dynamics of Sexting and Teen Social Media Cultures

**Amy Shields Dobson** [1,*] **and Maria Delaney** [2]

1    School of Media, Creative Arts, and Social Inquiry, Curtin University, Bentley, WA 6102, Australia
2    Centre for Resilient and Inclusive Societies, Deakin University, Geelong, VIC 3217, Australia;
     delaneymt@gmail.com
*    Correspondence: amy.dobson@curtin.edu.au

**Abstract:** This paper reports on data from interviews conducted with parents of high school-aged teens as part of a study which was aimed at better understanding the gendered dimensions of youth sexting and social media use, and the community responses to it. Here, we outline the findings on parents' key concerns around digital and social media, their perceptions of the gendered dynamics of youth self-imaging practices, and their attitudes towards sexting and potentially "sexualized" self-images. Echoing other research in this area, parents were not overly concerned about sexting, nor sexual image-sharing or sexual media use among teens. Rather, their key concerns were more generally about the intensities and pressures of constant contact with peers in the digital era. They did, however, articulate key gender differences and socialization processes around youth self-imaging practices. We discuss the gendered cultural "attunements to sexualization" that parents negotiate in relation to social media image-sharing practices and suggest that the perspectives and experiences described can be understood as part of a broader cultural "psychologization", and often psycho-pathologizing, of youth self-images in digitally networked intimate publics that is particularly intense around the vectors of gender and sexuality.

**Keywords:** sexting; social media; gender; sexuality; youth; sexualization; selfies; self-images; sex education

## 1. Introduction

This paper reports on data from 30 interviews conducted with parents of high school-aged teens, youth workers, and teachers as part of a larger project which aimed to better understand the gendered dimensions of youth sexting and social media use and the community responses to it. This project was designed in light of the research delving into youth sexting, evidencing the ways in which gender structures harms around sexting and image-based sexual abuse and harassment (IBSAH). The views and experiences of parents, youth workers, and teachers around gendered social constructs in youth cultures becomes important in this context. Here, we draw primarily on data from the twenty-two interviews conducted with parents of high school-aged teens, some of whom were also teachers, as distinct from the other eight interviews, with participants interviewed primarily in their roles as youth workers or teachers. We discuss key findings around: (1) parents' key concerns about digital and social media; (2) their perceptions of the gendered dynamics of teens' social media participation; (3) and relatedly, their perceptions and experiences around the gendered dynamics of self-image sharing for young people.

Research with families on digital media constitutes an important body of work on everyday life in the digital era. Research with young children, teens, and their parents/caregivers (hereafter "parents" to denote adults in significant caregiver roles) has examined the place of digital devices in family life, and how digital media is both shaping, and shaped by family life (Livingstone et al., 2011) [1]. Research has centrally detailed the difficult emotional tensions in relation to negotiations around use of digital devices (Clark,

2013) [2], the complex tensions between "care" and "surveillance" for parents (Balmford, Hjorth and Richardson, 2021) [3], and the intensity for young people of their "always on" social life through digitally connected devices (Turkle, 2011) [4]. The central importance of social class and cultural background in shaping families' attitudes and values in relation to digital technologies has been highlighted in the past (Clark, 2013) [2], especially in the context of globalization, neo-liberalism, and rapid social change (Livingstone and Blum, 2020) [5]. Parents' understanding and views on sexting and sexual media use have been explored less frequently (Barrense-Dias, Suris, and Akre, 2019) [6], and mostly in the context of quantitative research on young people's use of digital technologies through a frame of understanding digital "risks and opportunities" (Livingstone et al., 2011; Staksrud, 2021) [1,7].

Our central concerns are more distinctly around the social and material-discursive construction of gender in relation to digital technologies, influenced by critical feminist Foucauldian and new-materialist approaches to gender and technology (Evans and Riley, 2015; Warfield, 2017) [8,9]. Such work illuminates how gendered social structures and relations shape media technologies and are shaped by new technologies in co-constitutive ways. As such, digital sexual or body images, for instance, are not seen as a-priori "risks" for young people, but often become such, and unevenly so across inequitable social structures and intersecting identities through distinct, contingent, and repeated social processes and conditions (Ringrose and Eriksson Barajas, 2011; Dobson, 2015; van der Nagle and Tiidenberg, 2020) [10–12]. Furthermore, they become so through decisions that embodied raced, gendered, classed humans make about technology that embody race, gender, and class; centrally for issues of sex and gender, to enable and encourage data "promiscuity" across digital platforms (Chun, 2016) [13]. These feminist theories have informed the conceptualization of this research and our analysis of the data presented. In this project, parents, teachers, and youth workers were asked more specifically about the gendered dynamics of youth digital and social media cultures, their views on youth sexting, as well as on commercial kinds of sexual media content, and the social and cultural meanings of such. Echoing recent research on parents' views of youth sexual media practices (Staksrud, 2021) [7], the participants were not overly concerned about sexting, nor sexual image-sharing or media use among their teens, as was our stated focus; rather, their key concerns were more generally about the intensity of constant contact with their peers in the digital era. More uniquely in this study, parents articulated some of their perceived gendered differences and socialization processes around youth self-imaging for social media platforms. They also described their own and other adults' gendered "attunements" to signs of sexuality and desire in relation to the policing youth self-representation online. These findings contribute towards better understanding the social and cultural dimensions that shape youth digital cultures, sexual scripts, and occurrences of sexual abuse and harassment.

Below, we briefly summarize the recent history of debates and panics over the "sexualization of children", and the role of digital media within this, as we think this is key to understanding parents' readings of youth self-images in the contemporary cultural context. We then briefly define new digital forms of "image-based sexual abuse and harassment" and highlight this as a problem that echoes existing socially-structured patterns of violence and oppression. We outline methods of data collection and analysis before discussing the key findings mentioned above. We suggest that the experiences parents have described in this study can be understood as in relation to a broader cultural "psychologization" and psycho-sexual pathologizing of youth self-images in digitally networked intimate publics (Dobson, Robards, and Carah, 2018) [14]. That is, self-imaging is understood as linked to, and as a measure of, vernacular psychological notions of "self-esteem" and "self-value" in particularly important ways in the digital era; and, we suggest, that this is particularly intense around vectors of gender and sexuality and the reading of such into youth self-images.

## 2. Sexualization Panics and Digital Media

In the mid-to-late-2000s, news media stories, policy reports, and popular advice books around parenting, gender, and culture that claimed to document the "sexualization of childhood" in media and consumer culture became particularly prominent. Most of the research conducted on these panics over media and "sexualization" describe how, in a postfeminist, neoliberal, socio-political context, adults have been sensitized to cultural signs of—particularly—girls' and young women's' so-called "sexualization", and their implied cultural victimhood (Renold, Ringrose, and Egan, 2015) [15]. It has been noted that public discourses about childhood "sexualization" via media are classed and racialized, and scholars have mapped the ways in which such discourses implicitly construct a "universalized" white, middle-class child victim as the central subject of concern (Egan and Hawkes, 2009; Gill, 2012) [16,17], and unpacked the classed and racialized signifiers in specific objects of concern (Barker and Duschinsky, 2012; Payne, 2015) [18,19]. For instance, particular fashion items marketed to girls, including animal-skin prints, short skirts, and low-cut tops have been labelled as "skanky" (Oppliger, 2008) [20] and associated with sex-work or with low class status. In short, discourses of panic over "sexualization", while sometimes usefully highlighting issues of media objectification, gender stereotyping, and the commercialization of childhood (Barker and Duschinsky, 2012) [18], also contribute to an unhelpful cultural imaginary about youth sexuality and desire as tied to sexual victimhood, "low self-esteem", and general psycho-pathologization (Egan, 2013) [21].

Young people's use of social media has been a particular vector of concern in sexualization panics, with digital and social media often framed as both a key driver and signal of youth "sexualization" and cultural victimhood in gendered ways (Karaian, 2012, Dobson, 2015; Hasinoff, 2015) [11,22,23]. Critical humanities feminist scholars have argued that neither the sharing of digital self-images nor potentially sexualized self-images are simply and unquestionably "risks" or indicators of an assumedly negative "sexualization" for young people in the way in which some psychological scholarship implies (for instance, Staksrud, 2021) [7], but have become such through contingent social processes, as mentioned above. Research into the prevalence and gendered social and cultural dynamics of youth sexting and image-based sexual harassment and abuse (IBSHA) suggests that violence and harm are occurring along existing lines of sexual violence and harassment, where women, gender-diverse, and sexually diverse groups are disproportionately impacted. Within these demographics, people with disabilities, Indigenous people, black and brown people, and culturally and linguistically diverse groups are more likely to experience harassment and abuse (Henry, Flynn, and Powell, 2020; Andreasen, 2022). Henry, Flynn, and Powell (2020, pp. 1837–1838) refer to the taking, sharing, or distribution of nude or sexual images in non-consensual contexts as "image-based sexual abuse" (IBSA), and pestering people for sexual images, asking intimate questions, sending pornography or other sexual images in contexts where it is unexpected and unwanted as a form of "online sexual harassment" [24]. Ringrose, Regehr, and Milne, 2021 (among others) have found, for instance, increasing reports from young women receiving unwanted "dick pics" regularly from male peers, and sometimes strangers on Snapchat, and "photobombing" of such via Bluetooth [25]. They thus refer to "image-based sexual harassment and abuse" (IBSHA) together to capture a continuum of behaviors. This is distinct from earlier research (and still in some disciplines) that refers to "non-consensual sexting", "revenge porn", and "cyberbullying" to describe practices and behaviors that the scholars mentioned here have argued should be clearly labelled as forms of abuse, violence, and harassment: the term "revenge porn", for instance, unhelpfully eroticizes abusive behaviors, while "cyberbullying" depoliticizes a range of socially-structured abusive behaviors including sexism, racism, and homophobia (Dobson, 2019; Ringrose, Regehr, and Milne, 2021) [25,26]. In summary, research informed by feminist and social-justice oriented perspectives has highlighted the intensifications of existing lines of social oppression and violence, and new means of such that digital technologies sometimes open up; as distinct from less nuanced notions of digital technolo-

gies and social media driving youth "sexualization" and sexual "risks" or victimhood in universalized ways.

## 3. Method and Analysis

For this research, to better understand the gendered cultural dimensions of youth digital cultures and image-sharing practices, the views of parents were sought. Through semi-structured interviews, parents of school-aged teens were first asked about the devices and the key digital and social media platforms used by teens; rules and boundaries at home around digital communication and networked devices; sexting and cyber-safety education that they were aware of happening at schools; perceptions of the gendered dynamics of social media use among teens; views on gender representation in popular media; views on pornography and teens' use of it; views on youth sexting, including the possible benefits of digital sexual communication for teens; and how they would like schools to approach issues around sexual image sharing among teens. Participants were recruited through education and teaching research networks in South-East Queensland, including via advertising the research and seeking participants from an online network for feminist educators and parents, and snowballing for participants from the authors' existing networks. The cohort of participants was thus largely convenience-based. Interviews were conducted at locations that were convenient for the participants (which were most often at their homes or a local café) and were subsequently recorded and transcribed, with most being between 1–1.5 h in length. Participants were given a $50 gift voucher as a gesture of appreciation for their time. Ethical approval was received for the project from the University of Queensland. Participants have been anonymized with use of pseudonyms. The names of any friends of family members mentioned, and other identifying details have been changed to protect their privacy.

The networks from which we recruited mean that it this not a diverse nor representative cohort of participants, especially in regard to class or cultural background; but rather a highly educated, middle-class participant cohort. Most participants were female and working in highly skilled professional roles; only three parents were male. Social class, education, relationship status, and cultural background were not directly asked; rather, some information on this emerged in interviews. A small number of parents were immigrants to Australia; most were in heterosexual marriages; none explicitly identified as LGBTQI+ or gender-diverse, although a few mentioned children who did; and two mothers were currently single-parenting. Previous research has suggested that social class is key in shaping parents' values, approaches to, and experiences with digital media and technology (Clarke, 2013; Livingstone and Blum, 2020), and we suggest that it is also significant here in regard to the kind of liberal and feminist-oriented politics of most of our participants, and the particular ideas and discourses about gendered relations, sex, power, and equality that emerged in these discussions. The views that come across then, while not representative of diverse perspectives, may provide insights into a particular strand of heteronormative liberal discourse on gendered power relations in the postfeminist and digital era.

Both authors read through the interview transcripts to familiarize themselves with the data and coded manually in MS Word for the responses to interview questions, taking notes on the themes that they identified as being repeated/similar, and also on the common responses that they observed. We performed this separately before discussing our perceived repeated themes together to confirm and develop up from. Amy then wrote brief narrative-style summary notes for each interview, summarizing the interviewee's social and familial context, along with their key experiences, anecdotes, and views articulated. Both the coded transcripts and the collated summary notes were grouped together in order to identify key findings thematically across the data, and to identify key extracts from these transcripts (Braun and Clarke, 2006, p. 79), [27]. The data has been analyzed thematically, drawing on critical feminist Foucauldian perspectives to try to identify discourses and experiences issues of gender and power (Braun and Clarke, 2006), in line with our central concerns and feminist theoretical commitments mentioned above. Here we discuss

three key findings from participants' responses: (1) their articulation of the pressures and intensities around "always on" digital sociality for teens, and the "out of control" nature of digital life; (2) gendered socialization around social media practices, and the particular "attunement to sexualization" of girls that parents both enact and encounter from other adults; and (3) a suggested broader psychologization and psycho-sexual policing of youth self-images, which parents are navigating in distinctly gendered ways.

## 4. The Pressures and Intensities of "Always on" Sociality and Digital Life as "Out of Control"

Consistent with other studies on digital media and family life (Staksrud, 2021, p. 558; Clarke, 2013) [2,7], the constancy of digital media devices and communication itself, and thus the constant presence of peers through digital messaging platforms, was something parents stated as a key source of concern, more so than concerns over sexual media and communication. This was something that emerged in discussions of parents' key concerns, rather than a topic we set out to explore. Participants described how hard it was to control teens' engagement with their phones or other networked digital devices and a sense of powerlessness in this regard. Some parents articulated feeling frustrated, exasperated, out of control, and at a loss around the constant presence of networked devices, mostly phones. Others seemed to have accepted the loss in control around this as a condition of modern life for both their teens and themselves, making jokes about teens being glued to their phones, and about the phone being an extension of their teen's body or a "third arm". Several parents commented on the take-up of private group chat functions among teens and how they felt this intensified the impact of the phone's constant presence. Relatedly, several parents noted a shift away from family influence and towards peer influence and power, which they perceived to be further intensified by the functions of instant messaging and group chat. Fifi described this astutely. Her five sons ranged in age from 12–25 at the time of the interview. Reflecting on shifts she had witnessed over the years of her sons' lives, Fifi saw instant messaging as having facilitated a further shift away (beyond that often associated with this general life stage) from family and towards peers in terms of their constant presence, power, and influence in teens' lives, describing the impact of MSN in terms of bringing peers into the home as if they were "here all the time", and like having a constant "body of approbation" in the home attached to her teenage son.

Relatedly, several parents described their key concerns with digital media and communication as being linked with the intensity of pressure on their teens to be "always on" (Turkle, 2011) [4] socially, and thus to maintain a kind of Goffmanian "social face" even when in the home with no peers present, or at gatherings with only close family members. Charlotte's high-school aged sons were asked to turn off their networked devices two hours before bed, but Charlotte described this as a constant source of struggle. She perceived the pressure for her 15-year-old to be still chatting to his friends at night as immense. Similarly, Shone stated her key concerns about social media in terms of young people not getting a social "break" from their peers and having to be "always answerable". As she explained, "You might have had a terrible day and then you go home and you still have a terrible day [with peer conflicts potentially continuing], even into when normally you would be asleep [...]". Molly, a high school teacher and parent, noted that one of the key positives of social media and digital communication for teens is being "the place they go for support and help", but that this line of on-tap support was "very intense for kids". She stated, "their phones are ever sensitive and ever present should they need each other. That can be three o'clock in the morning. They've had conversations with me where they're [telling me about] helping each other out, getting each other through the night at three o'clock in the morning, four o'clock in the morning, five o'clock in the morning". As Molly also noted, the intensity of 24/7 contactability is a pressure and a source of anxiety and emotional intensity for adults too, who want to be contactable for their teens in case of emergencies, and struggle to balance this with the need to "switch off", connect with their partners, sleep, and generally "try not worry".

In summary, many parents communicated a sense that, whilst offering incredible benefits, opportunities, support, and fun, the place of digital media and communication in their lives felt pretty out of control. That is, living as part of an "always on" digitally networked public was clearly understood and explicitly framed by parents as not a choice—as the unchosen conditions of modern life that families are forced to navigate. One mother stated that she had considered the question of, if given a choice as to whether or not the Internet existed, despite all the benefits and good it offers, she would choose not to have it: she would, she told us, pull the plug on the Internet. The "unchosen-ness" of the broader techno-social conditions is important to keep in mind in relation to parents' negotiations around teens' use of social media.

## 5. Gendered Differences around Teen's Self-Imaging Practices on Social Media

Participants were then asked more specifically if they perceived gendered practices around social media use, either in the lives of their teens or more generally speaking. Several participants were hesitant to describe their experiences and key concerns around technology use by teens in gendered terms, pointing instead to their concerns around the place of digital communication in contemporary life more generally, as discussed above. Several of the participants described their teens or teens they knew as having non-gender-typical interests, and pointed to ways many teens' interests and online practices may be gender-subversive or not easily categorizable in binary-gendered terms. (No participants had teens identifying as gender-diverse at the time of interviews.) However, most participants noted the ways in which social media use and common practices with social media might happen in binary gender-typical ways. For instance, many parents noted that boys tended to be more interested in gaming and "informational and interest-based" YouTube videos on a range of topics, but centrally on sports and gaming; while girls were noted to be interested in Instagram and YouTube around appearance-related topics, such as fashion and beauty.

The gendered differences in social media use that parents tended to describe most often were around self-imaging practices. Participants discussed the gendered nature of self-image taking and sharing practices in relation to social media. It was noted by parents that boys were not socialized to want to share images of themselves publicly, whereas girls were; that boys did not post self-images frequently, that girls were socialized to want to share self-images, often in subtly sexualized ways, and that girls were also then using social media and self-image sharing more for self-validation and appearance validation purposes compared with boys: there was a general perception that girls may be using social media more for the "likes" compared with boys. For instance, Maree (mother of a teen boy) had noticed that "they don't share in the same way. They [boys] don't do the Snapchat and they don't do the—from what I've seen—they don't do all of those social media things in the same oversharing-way that girls do. They tend to use it on a 'well let's do it quietly under messenger and we'll just have a giggle amongst our mates', not out in the wider world". Carol, a mother to a teen girl and two boys in their 20s described what she had seen of boys' images compared with that of girls as follows: "No, their images, if they (her boys) put them up, they're photos which are—probably proportionately one is to ten, we're talking here—it will be about their sporting team, or the party they were at. Or the sculling competition at the local college". Esme (a mother of three teenage boys) commented on the gendered differences she had noticed, with her boys being less interested in image-sharing than the girls they were friends with. She said of her sons: "They don't want their photos on there. They don't like me posting photos. They don't post photos of themselves. They don't do a lot of selfies. The only photos I see are usually from parties, when there's been groups of people sharing photos, but they personally don't do that, whereas a lot of their friends who are girls, post a lot of photos on Facebook".

Girls' practices around sharing self-images were described as more frequent, more focused on self-portraiture, more sexualized via "pouts" and "cleavage here and there", and sometimes also as very "professional" looking. Carol noted that a lot of the self-images

she had seen shared by teen girls in her network looked almost "like airbrushed model poses and photos", which she felt was "a bit tragic", sounding sympathetic, and several other parents implied a similar sentiment in terms of noting the level of expertise and labor normalized for girls and young women to participate in social media image-sharing, and also in terms of the conformity with the now normalized celebrity and professional model standards. Esme discussed how for girls there was quite a high degree of pressure and normalization around image-sharing on social media, and she thought that because of this, social media use often became tied to self-esteem for girls in a way it often was not for boys. Other parents gestured towards similar views and were quite conscious of the social appearance-related pressures on girls, the general socialization towards "oversharing" for girls, and the ways in which social media attention, measured through likes and comments on self-images, became tied to validation and self-esteem in a way that it was not perceived to be for boys.

A few mothers of teen girls further discussed their negotiations and conversations around self-imaging practices, the implied "appropriate and inappropriate" in relation to such, and the perceived or implied link between self-images and self-esteem. In the following section, we describe three experiences related in interviews by mothers of teen girls that speak to what we describe as an "attunement" to signs of "sexualization" in relation to young women's self-images in particular; that is, a cultural sensitization to signs of sexualization in social media self-images shared by young women, which mothers both enacted and encountered from other adult viewers of teen's self-images.

## 6. Gendered "Attunement to Sexualization"

Many parents expressed their desired approach to allow teens to express themselves, to communicate with their friends, and use social media with some sense of freedom and privacy, rather than surveillance in the form of password sharing and constant monitoring. Those who did feel the need to practice a high level of surveillance were not happy about it and had well-justified contextualized reasons in terms of the age, maturity, and the individual developmental and psychological needs of their teen, including experiences of harassment and abuse by both peers and adults. Most participants did not want to set hard rules around self-representation that would involve telling teen girls in particular "how much is too much", as Carol put it, wanting to let teens' develop their own judgement about digital self-representation; and given that, as noted above, issues of self-imaging were described as important and sensitive matters for young women in particular. Participants often recounted more subtle forms of communication about notions of the "appropriate" and the "inappropriate", tried to keep their conversations light and friendly in tone, and to open up conversations with their teens about their self-representation online when opportunities arose for them to do so. Participants commonly expressed views that were very critical of sexist slut-shaming and victim-blaming discourses about women and girls' appearance. It was in this context that some mothers discussed feeling the need to try to unpack digital self-representation and sexual "reputation" explicitly with their daughters, to try to protect them from the imagined possible social and psychological harms that were seen as a "consequence" strongly tied to sexualized self-representation.

For instance, Karla (who had an older teenage boy and a young teen daughter) had been heavily policing both children's social media accounts and access to their devices since her son had been involved in a sexting incident a few years earlier. She sounded deeply conflicted about her surveillance practices in places, and she described feeling tired and drained by her efforts, stating: "It's so draining. I'm so sick of doing it. I am so sick of having to keep an eye on it". Her son had been involved in an incident where the police were brought into the school after a girl at her son's school sent around a topless image to a group of boys and asked them to reciprocate by sending something back. Her son sent something back in response, along with a few other boys, and when the school found out, the police were brough into the matter. The girl involved was suspended for a month, while the boys involved were each suspended for one day. Karla was really astonished and

angered by the discrepancy in the punishment. She thought it was unfair that the girl was suspended for substantially longer than the boys involved. She recalled serious discussions about this with her son following this incident; about the gendered roles at play here, the gendered double standards, and his own responsibility in this situation that, she said, he had wanted to deny. In the context of this experience with her son, Karla described the conversation she had with her daughter in relation to potentially sexualized self-images. She said: "I unfortunately had to have this conversation with my daughter. I unfortunately had to say to her if you, you know, "If this ever happens it's YOU that will get a reputation. It's so horrible to say that to her. And like I said before, in the other instance, it was the other way around". Karla was noting that, for her son, the conversation that followed the sexting incident opened an opportunity for her to discuss sexual double standards and try to responsibilize him in a way that aligned with her own values and ideals about gender equity. She seemed proud of the conversation she had with her son about this. In relation to her daughter, in contrast, Karla felt "horrible", as she stated—that is, obliged and very conflicted—about trying to responsibilize her daughter in relation to her self-imaging, and pre-emptively doing so, before anything "inappropriate" had actually come to her attention. In other words, what Karla felt was "horrible" here was the kind of reproduction and re-stabilization of gender and sexual double-standards she was participating in by having this conversation about imagined possible sexual self-images and her daughter's sexual "reputation". Having observed the gendered consequences of the sexting incident involving her son years earlier, Karla felt the need to raise this issue with her daughter in quite dire terms, out of a desire to protect her from gendered social consequences that felt immovable and out of her control.

In a different encounter with a cultural gendered "attunement to sexualization", Daisy recounted a situation where she had been alerted by a teacher from her daughter's school to an Instagram post featuring her daughter that another mother had perceived as inappropriate. She described how she had once received an urgently-toned phone call from a teacher calling to let her know that another parent at her daughter's school had complained that her daughter and her daughter's friend, aged 13 at the time, had posted images of themselves on Instagram in their "underwear". Daisy described her interpretation of the situation, which was that this was an overly-sensitized misreading of an image of the two friends in sportwear. Daisy recounted: "I got a phone call an hour later from the other mother, whose daughter was apparently in the 'underwear shots'. She said, 'I've looked. It's absolutely not—I don't know what they're on about. This is a complete outrage'. [...] It wasn't sexual, and they were wearing their sports gear. They were in bike pants and a top". Daisy went on to explain the conversations she had had with her teenage sons and daughters about their "digital footprints", and how her teen years before the Internet were completely different, stating: "I had the freedom to make mistakes, even with nakedness and sex, without public exposure, that they won't have because there can be video or things that can—and with their name attached to their social media accounts". Daisy was trying to foster trust and openness with all her children and described progressive and open conversations with them about gender, sex and power, consent, and feminism. She felt a need to emphasize the techno-social context of digitally networked publics that involve "digital footprints" and the possibility of a "complete loss of control" of one's data and images.

Other mothers discussed issues of digital self-representation with their daughters by trying to open up conversations about the possible meanings and cultural readings of self-images with them, as had Nicki and her husband, who were both teachers and parents to a young teenage girl and boy. They had only recently agreed to let their daughter Lizzi start an Instagram account on the condition that it was monitored by her parents, and they had access to her phone passcode. The rules were that if any "inappropriate" images were posted, the phone would be confiscated. Like Daisy, Karla, and several other participants, Nicki identified as a feminist, and discussed issues of gender and power at length, describing her views on the negative impacts of women's sexual objectification in

relation to clothing, images, and media. She had been keeping an eye on the Instagram posts and comments of friends' and peers in her daughter's network in relation to this newly permitted account and had been struck by the use of various sexualized emojis in comments on some posts, such as with tongues out and panting symbols, representations of wetness, and various fruits to symbolize body parts. Nicki gave a thoughtful and self-reflective account of a conversation she had with her daughter about such sexualized comments on an Instagram post by one her daughter's close female friends, Jen. After seeing comments about Jen's hotness and attractiveness in response to a selfie, Nicki tried to start a conversation with Lizzi about self-representation and, implicitly, self-sexualization, and the kind of comments Jen's selfie had garnered in response. Lizzi felt very confronted by these questions according to Nicki, in a way that took Nicki by surprise. Nicki recounted that Lizzi shut down and would not speak to her. Later that evening, Nicki went to make amends, and described her daughter as being in tears over her mother's line of questioning. As Nicki recounted:

> *"I just said to Lizzi, why do you think Jen has done this? Because there were 50 comments about, "you're so hot, you're so gorgeous, you're so awesome". She just sort of went—she closed down. "I don't know". Wouldn't look at me. "I don't know. I don't know". I said, you know, blah, blah, blah, blah, blah, and I'm just trying to think, you know? I said, you know Lizzi, I think if you posted a photo like this I'd be really disappointed and I would wonder why you're doing it. I don't really understand why you girls want to do it.*

> *She just would not answer. Then she didn't talk to me for the rest of the night. Then I sort of had a think about what I said, and I thought, you've got to be careful criticising—no teenager likes their friends being criticised. I remember that from being a teenager. You take it personally. I went in to give her a kiss goodnight, and I said, "aren't you talking to me yet?". She started crying and she said, "We just think that we look nice and we want to share it with our friends. It doesn't mean anything, Mum. It doesn't mean what you think it means.*

> *I went, "Okay". I think she was going down the track of you're taking this too seriously. You're reading too much into it. [...] But it was a real lesson for me. Because I just thought, okay, I've got to think about how we approach this".*

Nicki recounted her perception of her daughter's experience of apparently feeling quite confronted by her mother's "attunement to sexualization" in her friend's selfie and sexual attention in the comments on it. Nicki's surprise about this response could be due to her reading of signs of feminine "heterosexiness" (Dobson, 2015) [11] in the image perhaps being in line with the broader cultural ones typically associated with Jen's selfie, but being a reading that Lizzi perhaps either did not want to or was not ready to confront.

For Karla, the conversation about self-images and sexual reputation was one she felt that she "had to have" with her daughter, despite it being "horrible", and against her values of gender equity and anti-sexism in relation to sexual double standards. In a different encounter with the gendered cultural "attunements to sexualization", Daisy described both herself and the other mother involved being outraged by what they felt was an overly sexual misreading of their young daughters' Instagram image. Both Daisy and Karla were deeply concerned about whether they were doing the right thing or not; about whether the levels of trust and control they each enacted around social media would produce the desired outcome of a happy and safe child. Nicki's powerful account reflected on her teenage daughter's perspective of her mother's cultural "attunement" to the possible sexual meanings and audience reception of teen girls' normatively feminine selfies. Nicki was clearly surprised by how confronted and criticized her daughter felt in relation to her questions about self-imaging and the psychological motivation for such and was reconsidering her approach in relation to her account of her daughters' emotional response: "We just think we look nice and want to share it with our friends [...]. It doesn't mean what you think it means".

## 7. The Psychologization and Psycho-Sexual Pathologization of Youth Self-Images

In the accounts of parents discussed, we'd suggest we can see some ways in which public discourses around the "sexualization of children" perhaps shape and impact parents' negotiations and experiences of teens' digital and social media use, and how a broader cultural "attunement" to visual signs of feminine sexuality and desire plays out as something parents encounter and have to negotiate in relation to girls' and young women's social media images in particular, perhaps especially so given the white middle-class, professional, and feminist-oriented cohort of parents interviewed.

We suggest that the experiences and attitudes that these parents have described can also be understood in relation to a broader cultural "psychologization" of youth self-images. Byung-Chul Han's (2017) Foucauldian notion of "psycho-politics" is useful in this context [27]. Han's mobilization of Foucauldian approaches to power and subjectification in the digital era informs this analysis. Han builds on Foucault's ideas about power to conceptualize "psycho-politics" as distinct from "bio-politics" as the dominant form of power in subjectification processes in the form of neoliberalism distinct to the digital media era. In brief, his polemic account of life in the digital era develops the ideas of "psycho-politics" and "smart power" (2017, p. 15) to think through contemporary technologies and discourses of "emotional intelligence", "happiness", and "healing" as forms of psychological self-regulation that operate as internalized and "friendly" forms of power that are particularly intense, if not distinct, to the digital era. "Smart power", Han argues, is "friendly", and "Its signal and seal is the Like button". (2017, p. 15). If the disciplinary regimes of power were once concerned with, and manifest as, the organization of bodies, Han suggests, the neoliberal regime by contrast "seems like a soul", and "psycho-politics is its form of government" (p. 18) [28]. Psycho-politics is concerned less with the "discipline" of bodies and more with ideals of "motivation", "optimization", health, happiness, and "healing". Resonating with these ideas, we suggest that perhaps the parental and broader cultural "attunement" to signs of sexualization in young people's images that mothers of girls in particular described is related to the socio-political neoliberal logics of "psycho-politics" and "smart power" in the digital era. That is, vernacular psychological notions of "healthy self-esteem" and "self-value" have become prominent in digital media cultures and economies (Illouz, 2008). But further, such notions have come to be complexly tied up with self-imaging via increasingly and intensely visual social media platforms and cultures. Self-imaging has become widely understood as being linked to, and a measure of, self-esteem and self-value in this context. This happens for young people in particularly intense ways, structured by gender, race, and class; and visual signs of sexuality and desire are tied into various kinds of shaming along these socially structured lines (Dobson, 2015; Ringrose and Harvey, 2015; Payne, 2015; and Pitcan, Marwick and Boyd, 2018) [11,19,29,30]. Gendered cultural "attunements to sexualization" perhaps work within and as part of the broader logics of neoliberal, post-digital psycho-politics to produce a kind of psycho-sexual pathologization of youth self-images. That is, while many parents questioned the logics of slut-shaming and "sexualization", they all understood and were "working within" the broader cultural logics of a psychologized and sexualized pathologization of youth self-images, knowing that even subtle visual signifiers—such as a slight lip-pout or head-tilt—might be read by viewers of youth images as possible signifiers of sexuality or sexual desire. For girls in particular, broader cultural discourses around "sexualization" imply the public expression of any such signals as a "psychological issue" that needs attending to for the "optimization" and "healing" of souls.

To explain this notion of "psychologization" further, we want to try to unpack the apparently simple question Nicki posed to her daughter of why teen girls may take and share selfies that contain cultural signs of sexuality, sexiness, or desire. The "why" question was a question parents raised in interviews both humorously and in more serious tones about teens' self-imaging practices. It was also a common response given by parents when asked for their thoughts on sexting practices or potentially sexualized self-images among teens more generally. The conversation that Nicki described with her daughter about

her daughter's friend's selfie speaks to this broader normative cultural logic that we are particularly interested in that strives to unpack "the psychology" behind teen girls' images. This was present in several of the interviews conducted. Nicki asked her daughter *why* she thought her friend had done this; that is, why had her friend posted an image that had garnered lots of comments in response about her attractiveness and hotness. She told her daughter, "I think if you posted a photo like this I'd be really disappointed and I would wonder why you're doing it. I don't really understand why you girls want to do it". A response from Fifi about how she thought schools should be approaching youth sexting also echoed this kind of gendered and psychologized approach to youth self-images. In relation to what schools should be teaching about sexting, Fifi said, emphasizing the "why" question: "If it's a girl sending an image of herself to a boy, you have to then ask *why* she's doing that? I think for girls to be led to unpack the behavior, so more like—like kids should be taught psychology. They should be taught, okay, *why* would you do that?".

In brief, parents understood sexting, as well as potentially sexualized self-images, primarily as a "risk" in the techno-social context of digitally networked publics perceived as being completely out of control. Several participants suggested the approach of questioning "why" young people might take such risks as an approach that they understood as being distinct from gendered victim-shaming or blaming. In relation to sexting, parents mentioned the risks of "revenge porn" (Charlotte), being "terrified" of images falling into the wrong hands (Carol), the "what if" of images ending up on sites that victims of abuse don't have access to (Abby), the loss of control over images, and the permanence of images being out there "forever", and sexting thus being, as Molly put it "a black hole of all kinds of possibilities". This was, in places, articulated more explicitly in gendered terms, as discussed above, with girls suggested as needing to be made aware of the possibilities of lasting reputational damage, and that "what they do now will follow them", as Maree stated. But in response to these articulated gendered sets of "risks", there was also, we are suggesting, a related gendered "psychologization of self-images" often at work via this question of, given the risks, then "why".

Given the perceptions mentioned above that girls often came to use social media more readily and commonly than boys for self-validation through gaining "likes" on their images, parents were particularly thoughtful and sensitive about not wanting to judge, punish, or shame girls for displaying possible signs of "sexualization" that were either perceived by them, or imagined as a common possibility read into girls' self-images; instead, as mentioned, wanting to start open and curious conversations about the "why" and the "meaning" behind these images. This kind of approach was not distinct from but was often articulated alongside parents' feminist-oriented questioning of slut-shaming logics. While this urge towards such psychologically themed conversations appears to be understandably driven by the intention to not sexually shame teen girls, it also perhaps constitutes, and can be seen as evidence of, a kind of psycho-sexual, neoliberal, discursive normative logic whereby, for girls and young women (and we'd suggest more broadly for gender-non-conforming youth too), even a slightly "pouty" or slightly "too-posed" selfie may be questioned in terms of its implications about their psychological well-being, sexual desires, and the possible meanings they intend to signify to their audience. That is, it is perhaps the most culturally legible and neoliberal-logical response to suggest that girls be asked to psychologically "read into" their own self-images in terms of what they may indicate about their desires and motivations. However, in practice, as Nicki's account suggests, this can and is happening in ways that may be quite confronting for young people, and perhaps serves to further psycho-pathologize them around the vectors of sexuality and desire.

## 8. Conclusions

In this paper, we began by summarizing parents' key concerns and overall sense of digital media as feeling quite out of control, and of managing their own and teens' use of digital devices and social media as an unchosen condition of their modern life. This

characterization of everyday life in the digital era helps to contextualize the discussions that follow. As we outlined, participants were sometimes hesitant to describe teens' social media use in gendered terms, but many did note this. Self-image sharing practices were those that participants most notably described in gendered terms. We suggested ways in which broader cultural logics of "sexualization" shape parents perceptions and experiences of self-image sharing for young people as logics they both reproduce and question/push back against. In our Foucaldian-feminist influenced analysis, we have suggested a kind of gendered "psychologization of self-images" normatively at work in cultural logics around youth image sharing more broadly. The kind of psychologized readings of teens' self-images that we think has become increasingly normative and often taken for granted may be experienced by some young people as, in some ways, more intense and confronting than even highly intrusive rules and parental technological surveillance. That is, it is potentially very confronting for young people to be asked to consider their public self-representation at the level at which only public figures, such as celebrities and politicians, have had to in the past. However, this is now an everyday part of life in digitally networked "intimate publics" (Dobson et al.) [14], and one that adult authority figures have frequently emphasized for young people through responsibilization discourses. We are not saying that they should not do this. We are not saying that the responsibilization of youth around their own safety, privacy, and online self-presentation is a "bad" thing. We are trying to point to some of the everyday psychological intensities of life for both young people and those who care for them and about them that we perhaps often take for granted in digitally networked intimate publics. Moreover, we are attempting to point to the neoliberal logics of psycho-sexual pathologization and the policing of young people in gendered ways that everyday life in digital intimate publics has also perhaps come to normatively involve. As these interviews have shown, the mothers of girls in particular are encountering and negotiating these logics in quite complex ways. They are negotiating the hurt and confrontation that such logics may result in between teens and their parents, and they are caught up in them and by them, even as they also clearly question them and see them as "horrible".

**Author Contributions:** Conceptualization, A.S.D.; Methodology, A.S.D.; Validation, Maria Delaney; Formal analysis, A.S.D.; Data curation, M.D.; Writing—original draft, A.S.D.; Project administration, M.D. All authors have read and agreed to the published version of the manuscript.

**Funding:** This research was funded by a University of Queensland Postdoctoral Fellowship and the Advance Queensland Women's Academic Fund.

**Institutional Review Board Statement:** The study was conducted in accordance with the Declaration of Helsinki and approved by the University of Queensland Ethics Committee (approval number 2015001618).

**Informed Consent Statement:** Informed consent was obtained from all subjects involved in the study.

**Data Availability Statement:** The data are not publicly available due to ethical restrictions.

**Conflicts of Interest:** The authors declare no conflict of interest.

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
