# Peer review of "The “Psychologization” of Self-Images: Parents Views on the Gendered Dynamics of Sexting and Teen Social Media Cultures"

_2673-995X, doi:10.3390/youth3030063_

Round 1

Reviewer 1 Report

Really interesting and well considered research that places the semi-structured interviews into a theoretical context. It would be useful to have more detail about the how the cohort were recruited via the education and teaching networks and more details about the coding/identifying the themes. 

Other places that could be explored in more detail include some key concepts such as 'Goffmanian social face', self-esteem, self-value and shaming. 

The authors could also develop their neoliberal logic a little more in relation to how this relates to girls being asked to read into their self-images. 

There could be a limitation and recommendations section. 

Author Response

Response to review:

Thank you for this review.  The comments are appreciated. 

We’ve added a sentences in the methods sections to help clarify recruitment processes; as well as to clarify the process of identifying themes.

The other concepts mentioned for possible expansion are ones that are referenced but, given the word constraints and other reviews, there is not room to elaborate on these in further detail here. They have been discussed further in work by the authors and other scholars elsewhere. 

While we’ve mentioned limitations and recommendations in the text, including a specific section for this is tricky within the word limit and a bit outside our disciplinary conventions and preferred writing style, which adheres more to cultural studies conventions. 

Reviewer 2 Report

Thank you for the opportunity to read this insightful and interesting paper. It offers a much-needed nuanced examination of parents' perspectives and my reading of the paper suggests that a valuable contribution of the paper could be in articulating the tensions and ambivalence that characterises (predominantly self-described) mothers' experiences of addressing sexual self-imagery among girls. I offer some suggestions below that are hopefully of some use:

The literature review shows a solid understanding of the field and is succinctly summarised. However, there is a body of existing literature on parents' attitudes to and engagement with youth sexuality (including as sex educators in the home). It would be good to see some discussion of this literature regarding what is already known about parents' gendered narratives and orientations to risky youth sexualities (e.g., work by Davies, Elliot, etc.). 

Methods: there is a good explanation here of the methods/sample but it would also be good to know how the study was framed and how interviews were commenced. E.g., was the stated purpose on youth image sharing, or was there a more general framing? I say this because of the parents' concerns about constant connectivity etc. in general - was this spontaneously raised or purposively explored?

Findings:

- could the perceptions of the impacts of constant connectivity, particularly around the idea that parents are losing their teens to peers/devices, also be considered in terms of general processes of adolescent detachment and orientation to peers? Perhaps this process is being defined by these participants in terms of ubiquity of devices. 

- findings around the gendered dimensions of self-image on social media seemed to bring together data on parents' perceptions of their children and of young people (girls especially) in general. Perhaps this distinction could be made clearer, especially in light of evidence regarding how parents can 'other' matters like sexualisation. 

Near the bottom of page 9, I'd say it is necessary to make clear that any statements about Lizzi are based on Nikki's contributions in the interviews, not a fact as to how Lizzi reacted/behaved (currently it reads as the latter). There is also perhaps a critical point to be made here about how Lizzi's reluctance to engage may, as Nikki notes, be about the emotive framing of 'disappointment' by Nikki, which may be alienating. 

The arguments around psychologisation in the last section of the findings are quite unique, but some of the data in previous sections indicates that some parents' had a deep ambivalence about the gendered risks they constructed/perceived and the ways they discussed the ensuing issues with their children. There seems to be evidence of push back and the competing dimensions to parents' perspectives (within and between participants) could be examined a bit further. 

The conclusion could address some of the heterogeneity and ambivalence among parents and also, draw together some of the points around the intensity of always-on culture more generally (which did not seem gendered) and the specific focus on girls regarding sexual self-imagery. There could also be some noting of the nature of the sample - while the limitations are described in the methods, there could be some explicit consideration here of what the nature of the sample means for the data generated and directions for future research. Additionally, while mothers seemed to emphasise girls' sharing decisions and the risks there-of, less so was said, relatively, about abuse perpetrated by/toward girls beyond risks of distribution and shame/reputational damage (e.g., image based harassment and pressure to produce), which could be noted. Finally, a strong statement regarding the contribution to literature, with relevant citations (drawing on both the cited literature re image sharing and, ideally, wider literature on parents' attitudes to youth sexuality - as suggested above), is required.  

Minor grammar issues:

pg1 - is the acronym meant to be IBSHA?

pg1 - parent's and family's needs to be parents' and families' (I think). 

pg2, lines 55-60: long sentence with a lot of punctuation. It'd be easier to follow if broken up. 

pg2, line 78 - capitalised 'we' required at the start of the sentence. 

pg7, line 359 - 'responsibilize' (currently missing the first 'i')

The paper is generally well written with a good standard of English with some errors identified (see above)

Author Response

Many thanks for this review.  We’ve found these comments of use and appreciate them.

In regards to the suggestion about expanding the literature review to include work on parents’ attitudes towards youth sexuality more broadly, this is a good suggestion we will take up in future papers. However, the focus in the research and data reported here is on youth in the digital era, and our framing and approach has been on media and culture.  Adding a discussion of this broader work would require significant re-working and reframing of the paper, beyond the minor changes indicated by the editorial team.  It would also add more words than we have space to here, so we will kindly note this suggestion for future work.

We’ve clarified in the relevant section that parents’ concerns around general connectivity was something that emerged from discussions rather than a central theme of the research, and added sentences to clarify method details in the methods section.

We’ve added an edit to clarify that parents such as Fifi, quoted in findings, perceived the tech to have facilitated more peer influence beyond that associated with the general life-stage.

It is stated at the start of section 5 that this section brings together data on parents' perceptions of their children and teens more generally (in their experience).

We’ve edited to try to clarify in section 6 that any statements about Lizzi are based on Nikki's account in the interviews, not a fact as to how Lizzi reacted/behaved—many thanks for pointing this out.

We do think there is deep ambivalence and push-back from these parents and hope this comes through in the discussion of the data.  We’ve suggested here that attunements to sexualisation is something these parents encounter and have to deal with, not something they ascribe to uncritically; although most did revert to discussions of risk overall, despite feminist positions also articulated. We hope the edits made in concluding sections clarify this.   

We’ve re-written the conclusion to try to summarise the key points more clearly in response to reviewer comments. While we’ve mentioned limitations and recommendations in the text, including a specific section for this is tricky within the word limit and a bit outside our disciplinary conventions and preferred writing style.

Reviewer 3 Report

Overall, this paper was interesting to read as it provides a new perspective on how teens and their parents navigate life online. This paper could benefit from more commentary on the ontological and epistemological standpoints of their authors and how these inform their data analyses. Also, authors should remove expressions like "themes emerged" as the latter were more likely to be "identified" by the authors through their active engagement with the data. Also, I suggest that the authors include a reflexivity statement to inform the reader how their personal beliefs may have influenced data analysis and interpretation.

In the introduction, there is a statement that there were 30 interviews, but data are drawn from 22". Decisions with regard to what was analysed and why could have been fleshed out  more clearly in the method section (before results are presented) rather than in the introduction. I would suggest opening the paper with a statement of the issue/research question that needs our attention and justification for its importance.

With regard to results, it would be great if the authors described their sample a bit more accurately (if possible). Further, results would benefit from the inclusion of shorter quotes from the participants. I would also suggest significantly trimming down the quote on pages 8-9. When identifying themes, I believe each section should begin with a short definition/description of a given theme. Then, evidence and discussion should follow.

The quality of written expression is very good but there are some minor details that authors need to correct. For instance, in the abstract "Here, we outline findings on parent’s key" it should say "parents' key concerns". The manuscript would benefit from additional proofreading.

Author Response

Many thanks for this review.  We’ve found these comments of use and appreciate them.

We’ve tried to include some responses to clarify our epistemology and standpoint.  The description of material-discursive feminist work on page 2 states our epistemological commitments in brief, and we’ve added a line here to clarify that this informs our data analysis. We’ve also added lines in the methods section to clarify that this feminist theory and set of concerns has informed our thematic analysis of the data. We’ve edited the methods section in response to the comment here about “identifying” themes rather than claiming these as naturally “emergent”.  Appreciated!

We’ve added a sentences in the methods sections to help clarify recruitment processes, too.

The research problem is stated in the first paragraph, however, we’ve added a line to try to clarify this.

We’ve also edited to clarify in the intro that the 22 interviews analysed here were the ones with parents, as distinct from youth workers.

We’ve cut down the large quote in section 6 as suggested. However, given our disciplinary conventions and the paper style, we’re hesitant to cut down the quotes from participants as suggested by this reviewer, and to summarise each theme in concise terms, as the quotes and sections overall are geared towards providing readers with more contextualised narratives from the data.

We’ve proofed the manuscript on the whole in line with reviewer comments.